# Chronic Morphine Treatment and Antiretroviral Therapy Exacerbate HIV-Distal Sensory Peripheral Neuropathy and Induce Distinct Microbial Alterations in the HIV Tg26 Mouse Model

**DOI:** 10.3390/ijms25031569

**Published:** 2024-01-26

**Authors:** Danielle Antoine, Irina Chupikova, Richa Jalodia, Praveen Kumar Singh, Sabita Roy

**Affiliations:** 1Department of Surgery, Miller School of Medicine, University of Miami, Miami, FL 33136, USA; 2Department of Neuroscience, Miller School of Medicine, University of Miami, Miami, FL 33136, USA

**Keywords:** HIV-associated pain, neuropathy, opioid use, antiretroviral therapy, gut dysbiosis, Tg26 mouse model

## Abstract

Distal Sensory Peripheral Neuropathy (DSP) is a common complication in HIV-infected individuals, leading to chronic pain and reduced quality of life. Even with antiretroviral therapy (ART), DSP persists, often prompting the use of opioid analgesics, which can paradoxically worsen symptoms through opioid-induced microbial dysbiosis. This study employs the HIV Tg26 mouse model to investigate HIV-DSP development and assess gut microbiome changes in response to chronic morphine treatment and ART using 16S rRNA sequencing. Our results reveal that chronic morphine and ART exacerbate HIV-DSP in Tg26 mice, primarily through mechanical pain pathways. As the gut microbiome may be involved in chronic pain persistence, microbiome analysis indicated distinct bacterial community changes between WT and Tg26 mice as well as morphine- and ART-induced microbial changes in the Tg26 mice. This study reveals the Tg26 mouse model to be a relevant system that can help elucidate the pathogenic mechanisms of the opioid- and ART-induced exacerbation of HIV-associated pain. Our results shed light on the intricate interplay between HIV infection, ART, opioid use, and the gut microbiome in chronic pain development. They hold implications for understanding the mechanisms underlying HIV-associated pain and microbial dysbiosis, with potential for future research focused on prevention and treatment strategies.

## 1. Introduction

Distal Sensory Peripheral Neuropathy (DSP) is a prevalent neurologic complication that frequently arises in individuals afflicted with Human Immunodeficiency Virus (HIV). A significant source of morbidity [1,2], HIV-DSP is characterized by symptoms of neuropathic pain such as stabbing, numbness, and tingles [3] and afflicts at least 50% of individuals living with HIV [4]. While combination antiretroviral therapy (cART) has mitigated several HIV-associated complications, sensory neuropathies endure as a common neurological ailment linked to both HIV infection and its management with ART [5,6]. HIV-DSP arises from the degeneration and demise of distal axons and small unmyelinated sensory fibers, followed by the loss of larger myelinated ones [7]. Consequently, the primary etiology of pain in HIV-DSP is rooted in damage to the peripheral ascending nerves and central descending nerves. In particular, the excitation of damaged ascending sensory neurons underlies the heightened sensitivity to pain, resulting in sensations of burning or tingling [3,7]. 

Currently, no medications approved by the FDA specifically target the treatment of HIV-DSP. Clinical efforts have predominantly revolved around pharmaceutical interventions, including the use of opioids, to alleviate patients’ pain [8]. However, emerging data suggest that the chronic administration of opioid analgesics in individuals living with HIV/AIDS, particularly those undergoing ART, may paradoxically exacerbate chronic pain states [9,10] and contribute to increased adverse effects such as the disruption of the gut microbiome [11]. The gut microbiome encompasses trillions of microorganisms of different species and genes, such as bacteria, fungi, parasites, and viruses, that can be both helpful and harmful to the host’s body [12]. Over the years, these microorganisms have been revealed as key modulators of health and disease [12,13]. Notably, previous research from our lab using a rodent model [14,15] and from others using non-human primates [16] and human studies [17,18] have underscored the impact of acute and chronic opioid usage on the gut microbiome, which includes gut microbial dysbiosis characterized by compromised mucosal immunity, microbial translocation, and sustained systemic inflammation. During morphine treatment, beneficial bacteria decrease in abundance while pathogenic bacteria proliferate, resulting in a dysbiotic state that disrupts gut motility and gut barrier function [15,19]. Gut barrier dysfunction increases gut permeability, causing microbial translocation from the gut to other parts of the body and, consequentially, systemic inflammation [19]. Additionally, several gut microbial metabolites such as serotonin, acetylcholine, short chain fatty acids (SCFAs), and bile acids (BAs) can pass the blood–brain barrier and influence microglial activation and neuroinflammation [20,21,22]. Opioid treatment has been shown to alter the level of some of these critical metabolites such as SCFAs and BAs [15,16]. Opioid-induced gut dysbiosis resulting in sustained inflammation may perpetuate the development of HIV-DSP, as ongoing inflammation sensitizes nerves, thus making them more responsive to pain signals [23].

Importantly, HIV infection alone is associated with a dysregulation of the intestinal barrier and induction of gut microbial dysbiosis [24]. This dysbiosis has emerged as a pivotal contributor to the progression of HIV disease, as evidenced by the presence of microbial products in the peripheral blood of HIV-infected individuals, which has been implicated in immune activation and heightened morbidity and mortality [25,26]. Opioid use serves to exacerbate gut microbial dysbiosis in the context of HIV. Opioid administration in an HIV model of humanized mice leads to exacerbated microbial dysbiosis, gut barrier damage, and impaired intestinal renewal [27], and opioid abusers infected with HIV-1 showed elevated levels of microbial products, such as lipopolysaccharide (LPS), in their blood serum compared to non-users [28]. Consequently, in any investigation concerning HIV and its attendant complications, whether in the presence or absence of opioid utilization, the role of the gut microbiome demands consideration.

Several transgenic rodent models have been devised to simulate the impact of HIV infection with the aim of elucidating the pathogenesis of the disease. Among these models, the HIV Tg26 mouse has been designed to emulate contemporary HIV-1-infected individuals receiving ART [29]. Constructed utilizing non-infectious HIV-1 proviral DNA and possessing a C57BL/6 background, these mice serve as a valuable tool for investigating the neurological repercussions of HIV-1 infection [29,30]. Although this model is touted for its capacity to facilitate research into the neurological consequences of chronic, long-term HIV infection, behavioral assessments and the comprehensive characterization of its gut microbiome are at nascent stages of exploration. Data pertaining to neurocognitive function and sensory systems are only beginning to emerge [30,31].

Hence, in this study, our objective was twofold: First, we sought to evaluate the integrity of the nociceptive sensory system in the Tg26 mouse model and to investigate the progression of HIV-DSP within this model, particularly in the context of opioid administration and ART treatment. Second, we aimed to comprehensively profile the gut microbiome of the Tg26 mouse model, recognizing the significant role that microbial dysbiosis plays in HIV progression and its associated comorbidities. Though previous studies have investigated the relationship between morphine treatment, ART, and HIV-associated neuropathy, the novelty of this study lies in the use of the Tg26 mouse model. We hypothesized that opioid and ART interventions would exacerbate the development of HIV-DSP and to lead to distinctive alterations in the gut microbiota of the Tg26 mice when compared to their wild-type (WT) counterparts.

## 2. Results

### 2.1. Chronic Morphine Treatment Worsened the Development of Peripheral Neuropathy through Mechanical Pain Pathways in Tg26 Mice Compared to WT

We first examined body weight and nociceptive thresholds in Tg26 mice compared to WT to establish these responses in the context of HIV alone (Figure 1). Age-matched Tg26 mice weighted significantly less than WT (Figure 1a). When compared to WT, Tg26 mice displayed significant mechanical pain hypersensitivity and thermal hyperalgesia assessed with Von Frey filaments and the tail flick assay, respectively (Figure 1b,c). These results established that the expressed HIV proteins in Tg26 mice significantly impact the body weight and decreased mechanical and thermal pain thresholds. 

Next, to assess the nociceptive sensory system in the context of opioid use, Tg26 mice and WT mice were exposed to morphine at a concentration of 15 mg/kg or to saline for 10 days (Figure 2a). The results show morphine treatment to exacerbate weight loss (Figure 2b) and mechanical pain sensitivity (Figure 2c), in the WT and Tg26 mice. However, a significant decrease in thermal hyperalgesia (Figure 2d) was only observed in the WT mice and not in the Tg26 mice post-morphine treatment. 

### 2.2. ART Induced Peripheral Neuropathy through Mechanical Pain Pathways and Worsened Analgesic Tolerance to Morphine in Tg26 Mice

Accumulating evidence implicates ART treatment as a contributing factor in several HIV-induced comorbidities. Thus, to explore the role of ART treatment in the development of peripheral neuropathy in the context of HIV and opioid use, Tg26 mice were given ART through oral gavage once a day for a duration of 10 days along with the morphine treatment (Figure 3a). ART treatment alone had no impact on body weight (Figure 3b) but induced peripheral neuropathy by significantly decreasing mechanical pain thresholds (Figure 3c). Morphine treatment alone resulted in significant weight loss and mechanical pain hypersensitivity; however, the addition of ART did not further exacerbate these responses (Figure 3b,c). While morphine and ART treatments significantly decreased mechanical thresholds, they had no significant impact on thermal pain thresholds (Figure 3d) when compared to saline treatment. Consequently, these results show that morphine and ART treatments induce peripheral neuropathy mainly through mechanical pain pathways in the Tg26 mouse model. 

To further investigate the nociceptive sensory system of Tg26 mice and assess their response to repeated opioid exposure, the tolerance to morphine at a concentration of 3 mg/kg was evaluated in Tg26 and WT mice treated with either repeated 15 mg/kg doses of morphine alone (Figure 4a) or with ART (Figure 4c) for 10 consecutive days. WT and Tg26 mice showed a significant tolerance to morphine on day 11 compared to saline controls (Figure 4b). Similarly, following repeated morphine exposure along with ART, WT and Tg26 mice showed a significant tolerance to morphine on day 11 compared to saline controls (Figure 4d). However, Tg26 mice, but not WT mice, showed an exacerbated tolerance to morphine with ART treatment when compared to morphine treatment alone (Figure 4e,f). Thus, these results suggest ART treatment to exacerbate the development of morphine analgesic tolerance in the context of HIV.

### 2.3. Distinct Bacterial Communities Are Observed between WT and Tg26 Mice before and after Morphine Treatment

To understand the role of the gut microbiome in HIV-associated peripheral neuropathy and opioid analgesic tolerance, 16S rRNA analyses on colonic samples were performed in the various study groups. As in our previous findings [27], we did not observe a significant difference in the alpha and beta diversity between WT and Tg26 as shown by the Chao-1 index and Bray–Curtis PCoA plot between WT and Tg26 mice (Figure 5a,b). Further analysis showed a significant enrichment of bacteria belonging to family *Muribaculaceae* and a depletion of bacterial families *Marinifilaceae* and *Clostridiaceae* in the Tg26 mice compared to the WT control (Figure 5c, Appendix A). 

Morphine treatment did not cause any significant change in alpha diversity in WT and Tg26 mice (Appendix A) compared to the saline control. However, unlike WT mice, morphine treatment resulted in a significant shift in the beta diversity in Tg26 mice (Figure 5d,e). We observed a significant increase in phylum *Actinobacteriota* after morphine treatment in Tg26 mice but not in WT mice (Figure 5f,g). An LEfSe comparison between morphine-treated WT and Tg26 mice also highlighted several significant bacterial changes (Figure 5h). A family-level comparison showed increases in *Bacteroidaceae*, *Erysipelotrichaceae*, and *Sutterellaceae* families and a decrease in the *Lachnospiraceae* family in WT-morphine-treated mice compared to Tg26-morphine-treated mice (Appendix A). However, at the species level, only *Bacteroides acidifaciens* and *Bacteroides caecimuris* were significantly altered between morphine-treated WT and Tg26 mice (Appendix A). Additionally, a consistent decrease in *Eggerthellaceae* and *Ruminococcaceae* bacterial families and a consistent increase in the *Mycoplasmataceae* bacterial family were observed in both the WT and Tg26 groups after morphine treatment (Figure 5i). 

### 2.4. ART Treatment Modulates the Gut Microbiome in WT and Tg26 Mice

We next analyzed the 16S rRNA data from ART-treated WT and Tg26 mice. We observed a significant decrease in alpha diversity following ART treatment in WT mice but not in Tg26 mice (Figure 6a,b). However, the Bray–Curtis dissimilarity index showed overlapping bacterial communities between ART-treated WT (Figure 6a) and ART-treated Tg26 mice (Figure 6b) when compared to their respective saline controls. The *Rickenellaceae* family was significantly depleted in both WT and Tg26 mice after ART treatment (Figure 6c,e), while a significant increase in *Clostridiaceae* after ART treatment was observed only in the case of Tg26 mice. A groupwise comparison between ART-treated WT and Tg26 mice revealed a significantly lower alpha diversity assessed by both the Chao1 and Shannon indexes in ART-treated WT mice compared to ART-treated Tg26 mice (Appendix A), with no difference in the beta diversity (Appendix A). The LEfSe plot showed all the significant differences between WT and Tg26 mice after ART treatment (Figure 6f), including the *Muribaculum* and *Parasutterella* genera. 

## 3. Discussion

DSP is a well-documented condition that develops in individuals living with HIV, even in those undergoing ART [4]. DSP significantly impairs the quality of life for affected patients, primarily manifesting as persistent pain symptoms [4]. Consequently, patients with HIV often require pain relief medications, including opioids. Paradoxically, opioid use can exacerbate pain symptoms and contribute to disease progression through opioid-induced microbial dysbiosis [9,11]. Gut dysbiosis results in sustained inflammation [19], which may exacerbate HIV-DSP development via neuronal sensitization [23]. Our study aimed to investigate the development of HIV-DSP and assess alterations in the gut microbiota, particularly in the context of opioid use and ART treatment, using the Tg26 HIV mouse model. Though characterization is still in its nascent stages, the Tg26 mouse model has been employed to study the pathogenesis of HIV infection [29,30]. In our present study, both morphine and ART treatments not only exacerbated the development of HIV-DSP in the Tg26 mouse model, primarily through mechanical pain pathways, but also led to distinctive microbial changes when compared to WT mice.

In the context of HIV alone, our data indicated that Tg26 mice develop HIV-DSP, characterized by mechanical and thermal pain hypersensitivity when compared to WT mice. Similarly, HIV-DSP development has been observed in other HIV mouse models, including those infused with the HIV-1 coat protein Gp120 [32,33], resulting in both mechanical and thermal hypersensitivity, and those infused with the HIV-1 Tat protein [34], leading to mechanical pain hypersensitivity. It is noteworthy that several reports showed no changes in thermal hyperalgesia in the HIV-1 Tat model [34,35,36]. Overall, these observations of HIV-DSP development in mouse models are consistent with numerous reports in HIV patients [4]. For example, a study involving 1044 people living with HIV found that over half of the participants experienced HIV-DSP, with 38% reporting painful distal neuropathies and 24% experiencing non-painful symptoms [37].

Since many HIV patients, including those on ART, experience persistent pain, opioids are often prescribed. In the context of opioid treatment, our data revealed that Tg26 mice exhibited increased weight loss and mechanical pain hypersensitivity, although thermal hyperalgesia remained unaffected, following seven days of chronic morphine treatment, with or without ART. Consistent with this work, a previous study using an HIV Gp120 mouse model demonstrated that repeated morphine administration led to increased hypersensitivity to mechanical stimulation [38]. In our study, although ART did not exacerbate opioid-induced weight loss and mechanical pain hypersensitivity, it did worsen the tolerance to morphine in the Tg26 mice compared to WT mice. Morphine tolerance is a primary factor leading to reduced pain control, necessitating dose escalation and thereby making pain management more challenging [39,40]. Heightened morphine tolerance results in increased opioid use and magnifies the severity of related side effects. Overall, our results in the Tg26 mouse model align with the findings of clinical studies, indicating that the chronic use of opioid analgesics exacerbates pain in HIV patients [41,42].

We extended our investigation of the Tg26 mouse model to include the gut microbiome, given that opioid-induced microbial dysbiosis has been implicated in HIV-1 disease progression [43]. Our data revealed distinct differences in the bacterial communities between WT and Tg26 mice both before and after morphine treatment. Before morphine treatment, Tg26 mice exhibited an enrichment of bacteria belonging to the *Muribaculaceae* family and a depletion of those belonging to the *Clostridiaceae* family compared to WT controls. Similarly, a previous study found that the HIV Tat expression in 12-month-old female mice caused severe dysbiosis, characterized by a significant increase in *Muribaculaceae* and a decrease in *Lachnospiraceae* and *Ruminococcaceae* [44]. Following opioid use, our data showed morphine treatment to induce a significant shift in the bacterial community in Tg26 mice, as indicated by the beta diversity, leading to a substantial increase in the *Actinobacteriota* phylum compared to saline treatment. When compared to WT mice, at the genus level, morphine treatment in the Tg26 mice resulted in a decrease in *Dubosiella and Prevotellaceae,* which have been correlated with an anti-inflammatory response in the gut in the context of dextran-sulfate-sodium-induced colitis [45]. Additionally, morphine treatment resulted in a significant increase in the genera *Lachnospiraceae*, *Candidatus Saccharimonas*, and *Rikenellaceae*. Specifically, the increased abundance of *Lachnospiraceae* has been linked to potentially triggering an excessive immune response and intestinal inflammation [46], while *Candidatus Saccharimonas* has been denoted as a pathogenic bacterium [47]. The expansion of *Candidatus Saccharimonas* has been linked to inflammatory diseases such as gingivitis and other periodontal dysfunctions [48]. In summary, our data demonstrate that morphine treatment results in distinctive dysbiotic microbial changes in the Tg26 mice compared to WT mice, consistent with our earlier findings indicating that morphine treatment induces gut microbial dysbiosis and impairs intestinal epithelial repair in the Tg26 mouse model [27].

Furthermore, we observed that ART treatment led to a decrease in alpha diversity in WT mice but not in Tg26 mice. Recent studies involving HIV-infected individuals have reported a decrease in alpha diversity following ART treatment [49,50,51]. ART also significantly depleted the *Rickenellaceae* family in both WT and Tg26 mice, while significantly increasing the *Clostridiaceae* family in the Tg26 mice. Increased bacterial species within the *Clostridiaceae* family have been associated with systemic levels of bacterial translocation and inflammation, which have been successfully reduced in HIV patients through probiotic treatment with *Saccharomyces boulardii* [52]. Additionally, the *Rikenellaceae* RC9 gut group, belonging to the *Rikenellaceae* family, plays an essential role in crude fiber digestion [53,54] and is depleted in HIV patients receiving ART, leading to a reduced butyrate synthesis and disrupted metabolism in the context of obesity [55]. Moreover, previous studies investigating markers of microbial translocation and systemic inflammation in HIV patients receiving ART also found the *Rickenellaceae* family to be significantly depleted in the HIV-positive patients compared to HIV-negative controls [56,57]. Consequently, our results in the Tg26 mice are consistent with findings that patients living with HIV and receiving ART exhibit distinct microbial alterations, potentially associated with sustained systemic inflammation and disease progression. Overall, opioid- and ART-induced gut microbial dysbiosis, leading to increased inflammation, may underlie the exacerbation in the development and persistence of HIV-associated pain in patients.

Limitations exist in our study, necessitating further investigation. Our study primarily focused on the impact of morphine and ART on the development of HIV-DSP in the Tg26 mouse model and the characterization of its gut microbiome. As a result, interventions aimed at preventing HIV-DSP development were out of the scope of this study and were not explored. Future work will delve into treatments designed to prevent or mitigate HIV-DSP development. These studies will help to further elucidate the correlation between morphine treatment, ART, and HIV-DSP development. As the Tg26 mouse model’s microbiome is better characterized, future research will examine the direct role of the gut microbiome in HIV-DSP development and investigate whether probiotic interventions aimed at restoring a healthy gut microbiome can attenuate HIV-DSP development. 

## 4. Materials and Methods

### 4.1. Animals

All animal care and procedures were approved by the Institutional Animal Care and Use Committee at the University of Miami to ensure ethical treatment of animals. The animal subjects used in this study were 4–6-month-old adult male transgenic HIV Tg26 mice of C57BL/6 background [58] and age-matched littermate wild-type (WT) mice. Given the nature of this study involving non-human animals, a formal consent procedure, as typically applied in human studies, was not applicable. The transgenic Tg26 mouse expresses a 7.4 kb transgene containing the genetic sequence for the HIV-1 tat, env, rev, nef, vif, vpr, and vpu genes, which are transcriptionally regulated by the long terminal repeat promoter [59]. All mice were genotyped from tail biopsies collected at the time of weaning. All animals were housed two to five per cage in SPF conditions with ad libitum access to food and water.

### 4.2. Animal Treatments

#### 4.2.1. Morphine Treatment

Morphine sulfate obtained from the National Institutes of Health [NIH]/National Institute on Drug Abuse [NIDA] located in Bethesda, MD, USA, was dissolved in sterile saline. Mice received a constant dose of 15 mg/kg of morphine or saline intraperitoneal (IP) injection twice a day for a duration of 10 days. Nociceptive response was evaluated on days 0 and 11 after morphine treatment. To evaluate analgesic tolerance to morphine, the same morphine regiment was conducted in which mice received a constant dose of 15 mg/kg of morphine or saline IP injection twice a day for a duration of 10 days. On day 11, analgesic tolerance to a test dose of morphine (3 mg/kg) was evaluated.

#### 4.2.2. ART Treatment

The antiretroviral treatment (ART) included elvitegravir 30 mg/kg (integrase inhibitor), tenofovir 50 mg/kg (nucleotide reverse transcriptase inhibitor), and emtricitabine 30 mg/kg (nucleotide reverse transcriptase inhibitor) dissolved in sterile saline. Mice received the ART through oral gavage (200 mL per mouse) once a day for a duration of 10 days.

### 4.3. Nociceptive Testing

Pain-like behavior and nociception were assessed with the tail flick assay to evaluate thermal pain thresholds and the Von Frey assay to evaluate mechanical pain thresholds. 

#### 4.3.1. Tail Flick

During the tail flick assay, rodents were loosely restrained, and heat stimulus was applied to their tails. The radiant heat stimulus, emitted by a high-intensity lamp, was directed at the mid-region of the tail, with a maximum cutoff of 10 seconds to minimize tissue damage. The time taken for the tail to “flick” or twitch was recorded. The latency to tail-flick was determined in one trial.

#### 4.3.2. Von Frey

During the Von Frey assay, animals were placed individually in small cages with a mesh bottom. The “ascending stimulus” method was used to determine the mechanical paw withdrawal threshold. A monofilament was applied perpendicularly to the plantar surface of the hind paw until it bent, delivering a pre-determined force from 0.4 g to 11 g for 2–5 seconds. A response was considered positive if the animal exhibited any nociceptive behaviors, including brisk paw withdrawal, licking, or shaking of the paw, either during application of the stimulus or immediately after the filament was removed [60].

### 4.4. 16S rRNA Gene Sequencing

To evaluate the gut microbiome, the contents of the colon were collected under aseptic conditions from all mice after sacrificing. Sequencing was performed by the University of Minnesota Genomic Center, MN. DNA was isolated using DNeasy PowerSoil^®^ kits (Qiagen, Germantown, MD, USA) modified to include a bead-beating step.

### 4.5. Microbiome Sequencing Data Analysis

For analysis of the microbiome sequencing data, the Bioconductor platform with a specific emphasis on the dada2 version (3.17) package was employed [61]. The dada2 pipeline provides a comprehensive platform tailored for high-resolution analysis of amplicon sequencing data, such as the 16S rRNA gene data that are pivotal in microbiome studies. A quality profile check was performed using the plotQualityProfile function. Raw reads were then subjected to filtering and trimming based on criteria including truncation length, maximum expected errors, and quality score. Post filtering, the error rates for both forward and reverse reads were estimated. Subsequent steps included dereplication to collapse identical sequences, denoising to model and correct amplicon errors, and the merging of paired reads. A sequence table was constructed to tabulate the frequency of each sequence variant in the samples. Notably, potential chimeras were identified and removed. For taxonomic assignment, the Silva database, specifically the silva_nr99_v138.1_wSpecies_train_set.fa.gz reference, was utilized to match our sequences [62,63]. 

### 4.6. Statistical Analysis and Diversity Metrics

All behavioral data were analyzed with a student’s *t* test or a one-way ANOVA for multiple-group analysis using PRISM. For the microbiome data, the LEfSe (Linear Discriminant Analysis Effect Size) was used as it is adept at identifying taxa that are statistically different between groups, was used [64]. The threshold for considering differential abundance was set at an LDA score of 2, ensuring robustness in our findings. In assessing diversity, the *t*-test was employed for alpha diversity comparisons. Beta diversity, representing differences between microbial communities, was evaluated using the PERMANOVA test, a non-parametric multivariate analysis of variance. For multiple testing corrections in our *t*-tests, the Bonferroni method was used to ensure stringency in the *p*-value adjustments.

## 5. Conclusions

Our research highlights the interaction between chronic morphine treatment, ART, the development of HIV-associated chronic pain, and alterations in the gut microbiome using the Tg26 mouse model. The findings demonstrate that both morphine and ART exacerbate the development of HIV-DSP, resulting in increased pain sensitivity, while also inducing distinct changes in the gut microbiota. These insights are of significant clinical importance, given the challenges that HIV-DSP presents to affected individuals. While our study has made substantial progress in understanding these interactions, it also points to the need for further investigation, including the exploration of preventative measures and potential interventions. Future research will delve into treatments that can mitigate HIV-DSP development and directly assess the role of the gut microbiome in this process. Overall, our work emphasizes the value of the Tg26 mouse model in unraveling the complex mechanisms behind the opioid- and ART-induced exacerbation of HIV-associated pain and the implications of opioid- and ART-induced gut microbial dysbiosis. 

## Figures and Tables

**Figure 1 ijms-25-01569-f001:**
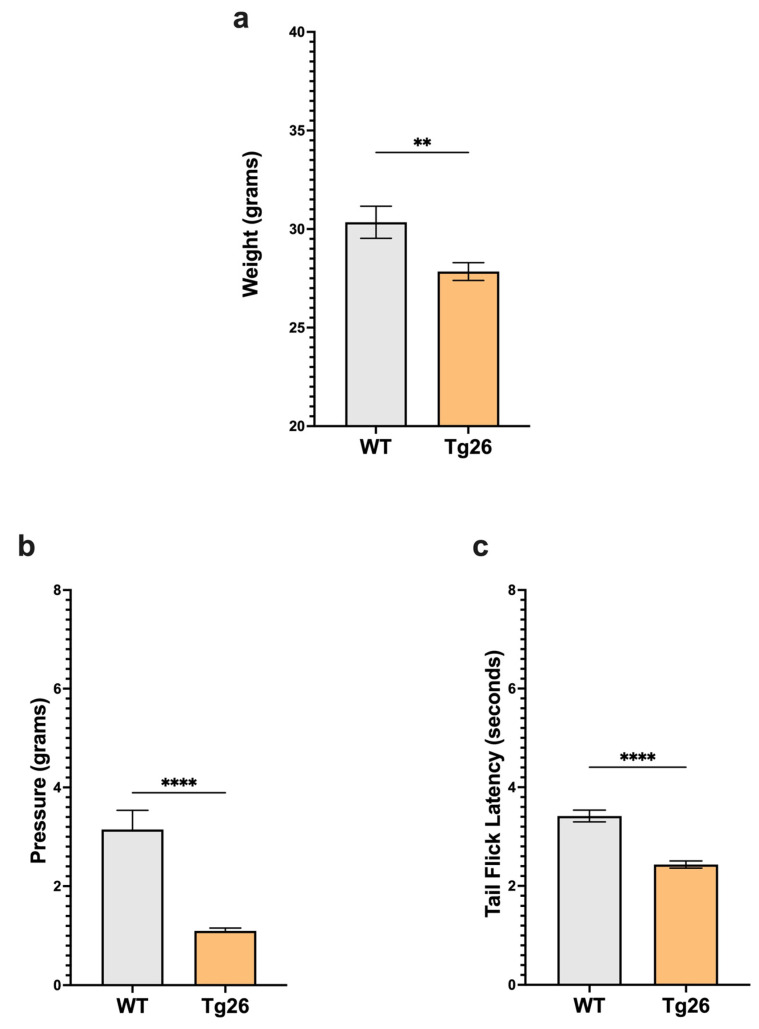
Tg26 mice showed significant weight loss and mechanical and thermal pain hypersensivity compared to WT mice. (**a**) Tg26 mice show reduced weight compared to WT. (**b**) Tg26 mice show significantly lower mechanical thresholds and (**c**) significantly lower thermal thresholds when compared to WT (n = 32 per group). Data were analyzed with a student’s *t* test and are expressed as mean +/− SEM (** *p* < 0.01; **** *p* < 0.0001).

**Figure 2 ijms-25-01569-f002:**
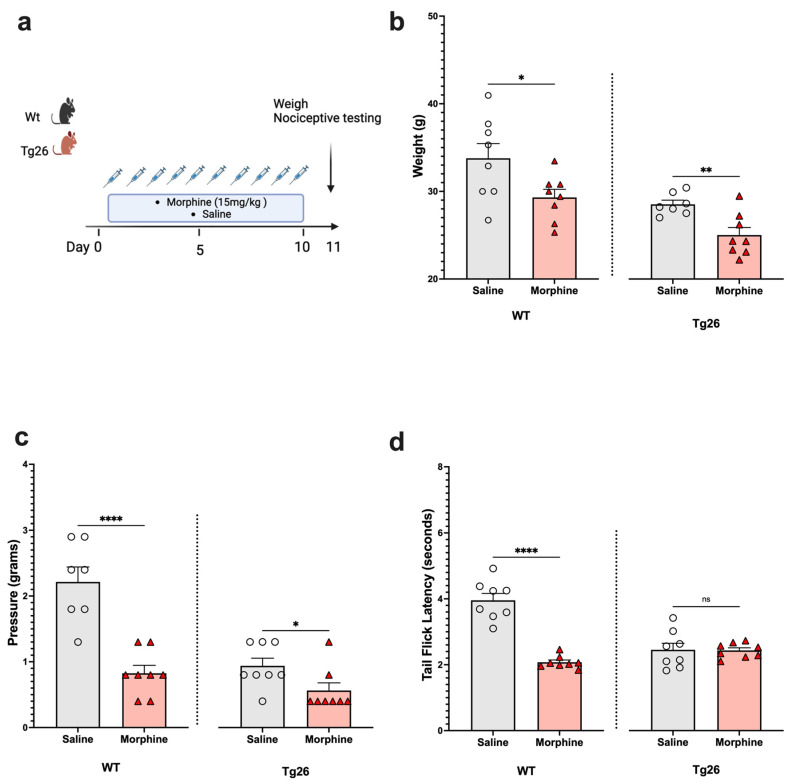
Morphine treatment exacerbated weight loss and mechanical pain hypersensitivity, but not thermal hyperalgesia in Tg26 mice when compared to WT mice. (**a**) WT and Tg26 mice received a constant dose of 15 mg/kg of morphine or saline intraperitoneal injections twice a day for 10 days. (**b**) Morphine treatment resulted in significant weight loss in both WT and Tg26 mice. (**c**) Morphine treatment resulted in significantly lower mechanical thresholds in both WT and Tg26 mice. (**d**) Morphine treatment resulted in significantly lower thermal thresholds in WT but not in Tg26 mice (n = 7–8 per group). Data were analyzed with a student’s *t* test and are expressed as mean +/− SEM (* *p* <0.05, ** *p* < 0.01, **** *p* < 0.0001; ns: non-significant).

**Figure 3 ijms-25-01569-f003:**
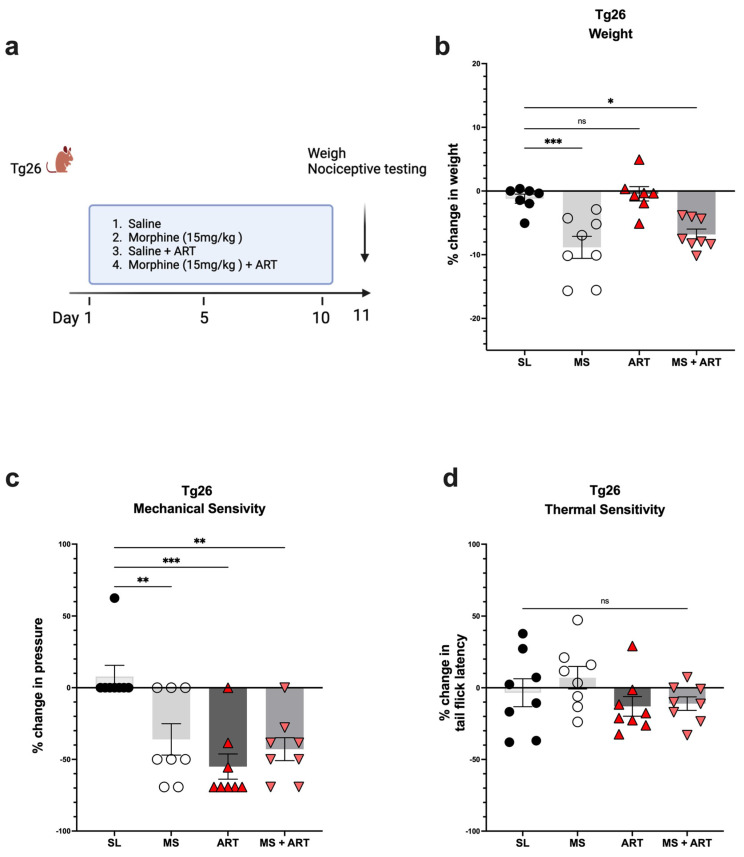
Morphine and ART treatments significantly decreased weight and mechanical pain thresholds, but not thermal pain thresholds in Tg26 mice. (**a**) In addition to the chronic morphine treatment, Tg26 mice were given ART through oral gavage once a day for a duration of 10 days. (**b**) Morphine alone and Morphine + ART, but not ART alone, resulted in a significant decrease in body weight when compared to the saline treatment. (**c**) Morphine alone, ART alone, and Morphine + ART resulted in a significant decrease in mechanical pain thresholds (**d**) but not thermal pain thresholds when compared to the saline treatment in Tg26 mice. (n = 7–8 per group) Data were analyzed with a one-way ANOVA and are expressed as mean +/− SEM (* *p* <0.05, ** *p* <0.01, *** *p* < 0.001; ns: non-significant). SL: Saline, MS; Morphine, ART: Antiretroviral.

**Figure 4 ijms-25-01569-f004:**
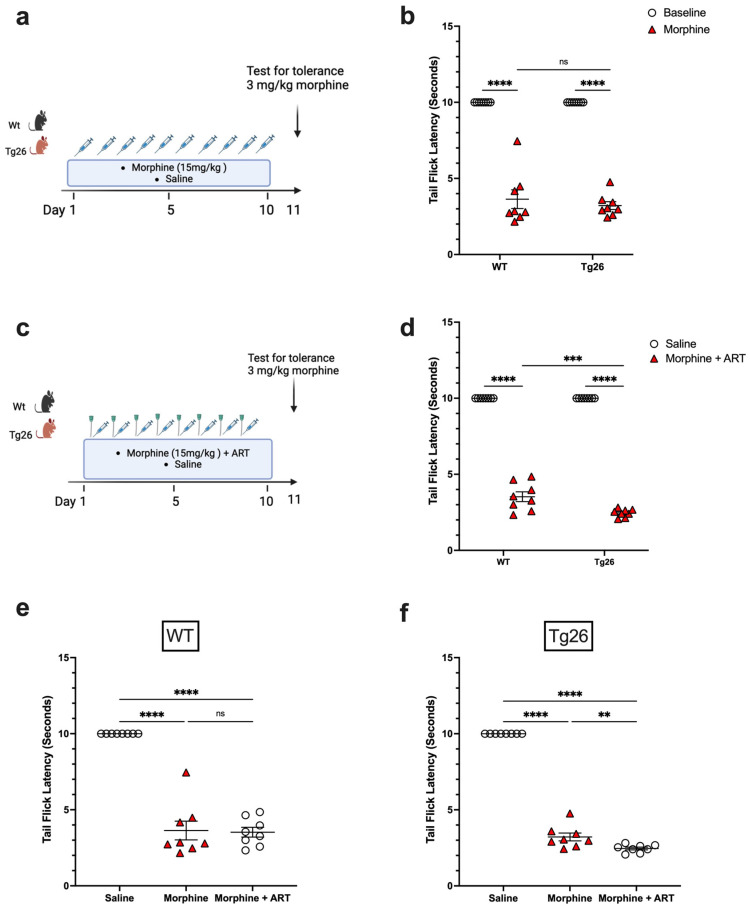
ART treatment exacerbated tolerance to morphine in Tg26 but not in WT mice. (**a**) WT and Tg26 mice received a constant dose of 15 mg/kg of morphine or saline intraperitoneal injections twice a day for 10 days and tolerance to 3 mg/kg morphine was tested on day 11. (**b**) WT and Tg26 mice showed significant tolerance to morphine on day 11 compared to saline controls. (**c**) In addition to morphine treatment, WT and Tg26 mice received the antiviral treatment (ART) through oral gavage and tolerance to 3 mg/kg morphine was tested on day 11. (**d**) On day 11, WT and Tg26 mice exposed to both morphine and ART showed significant tolerance to morphine compared to saline controls. (**e**,**f**) Tg26 mice, but not WT mice, showed exacerbated tolerance to morphine with ART treatment (n = 8 per group). Data were analyzed using a 2-way ANOVA and are expressed as mean +/− SEM (** *p* <0.01, *** *p* < 0.001, **** *p* < 0.0001; ns: non-significant).

**Figure 5 ijms-25-01569-f005:**
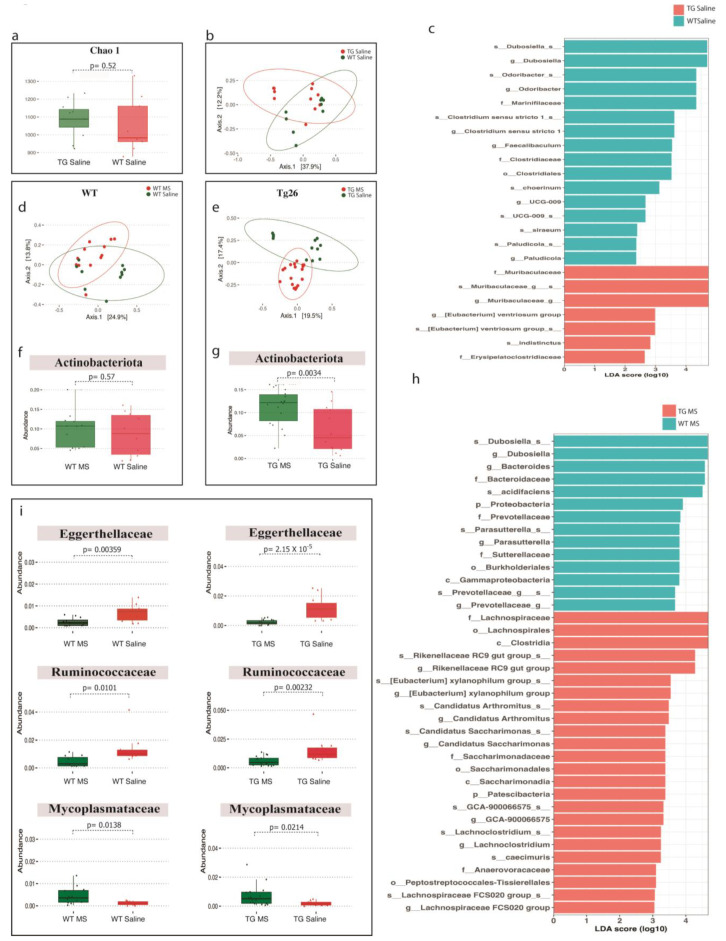
Distinct bacterial communities were observed between WT and Tg26 mice before and after morphine treatment. (**a**) Chao-1 index showing alpha diversity and (**b**) PCoA plot using Bray–Curtis showing beta diversity differences between WT and Tg26 mice microbiomes (q < 0.05). (**c**) LEfSe plot showing phylum differences between WT and Tg26 mice. Compared to saline treatment, morphine treatment showed (**d**) overlapping microbial communities in WT (q > 0.05), and (**e**) a significant shift in Tg26 mice after morphine treatment (q < 0.05). Morphine treatment in (**f**) WT mice did not change phylum *Actinobacteriota*, but (**g**) led to significant enrichment in Tg26 mice. (**h**) LEfSe plot showing phylum differences between morphine-treated WT and Tg26 mice. (**i**) Morphine caused consistent changes in bacterial families in WT and Tg26 mice (n = 7–8 per group). Alpha diversity and abundance between the groups were analyzed using the student’s *t* test and beta diversity was analyzed using the PERMANOVA.

**Figure 6 ijms-25-01569-f006:**
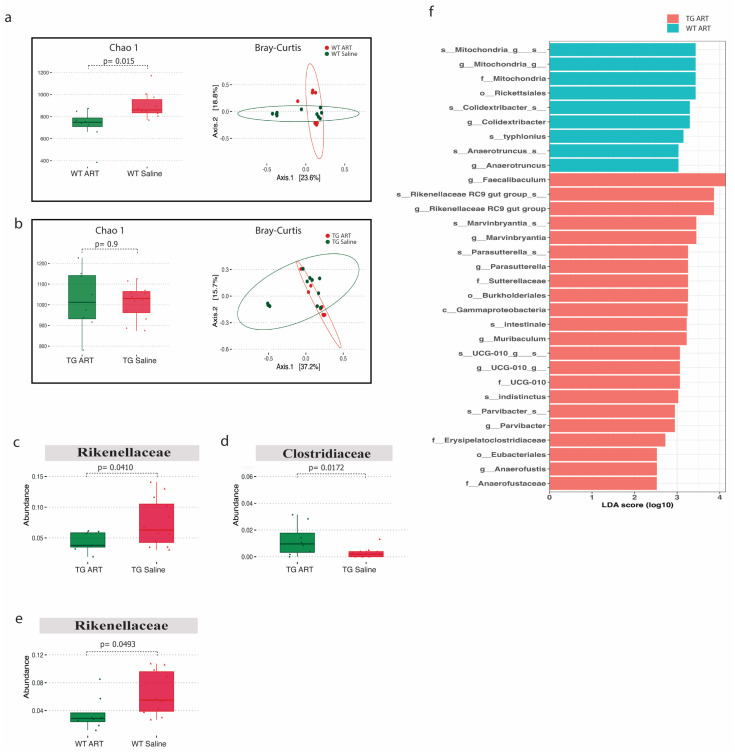
ART treatment modulated the gut microbiome in WT and Tg26 mice. (**a**) Alpha diversity assessed with the Chao1 index significantly decreased in ART-treated WT mice and PCoA plot using Bray-Curtis assessing beta diversity showed overlap in bacterial communities (q > 0.05). (**b**) No distinct change in alpha and beta diversity (q > 0.05) was observed in Tg26 mice after ART treatment. ART treatment in Tg26 mice caused a significant (**c**) decrease in *Rickenellaceae* family and (**d**) increase in *Clostridiaceae* family. (**e**) ART treatment in WT mice caused a significant decrease in *Rickenellaceae* family. (**f**) LEfSe plot showing all significant differences between ART-treated WT and Tg26 mice (n = 7–8 per group). Alpha diversity and abundance between the groups were analyzed using the student’s *t* test and beta diversity was analyzed using the PERMANOVA.

## Data Availability

All data supporting the findings of this study are available within the article and its Appendix A.

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
