# Peer review of "Chronic Morphine Treatment and Antiretroviral Therapy Exacerbate HIV-Distal Sensory Peripheral Neuropathy and Induce Distinct Microbial Alterations in the HIV Tg26 Mouse Model"

_ijms, 2024, doi:10.3390/ijms25031569_

Round 1

Reviewer 1 Report

Comments and Suggestions for Authors

The authors have used Tg26 mouse model to study the relation between Morphine treatment, Antiretroviral therapy and HIV-Distal Sensory Peripheral Neuropathy. They also report distinct bacterial community  WT and Tg26 mice before and after morphine treatment. More detail needs to be given to link gut microbiome with the morphine treatment.

The study has been conducted nicely and proper controls have been used. Though several reports are available depicting the relation between morphine treatment, ART and HIV associated neuropathy, the exception here is the use of Tg26 mice model. It would be nice if authors can address the novelty of the study in the introduction section. 

As the authors have mentioned in the discussion, interventions aimed at preventing HIV-DSP development were not explored. Such study would actually help in establishing the real correlation between morphine treatment, ART and HIV-DSP.

The figures are blurred at many places. The quality needs to be improved.

The authors are suggested to include a little more information on gut microbiome in the context of HIV infection and how is it linked to or hypothesized to be linked with HIV-DSP.

Reviewer 2 Report

Comments and Suggestions for Authors

The authors present an extensive where they investigated the development of HIV-DSP using a murine HIV Tg26 model through the administration of morphine and ART and further assessed the change in the gut microbiome in response to these treatments.

The paper is well structured and the results are presented clearly. Subject to a few minor revisions listed below, the paper is suitable for publication:

·         Figures 5, 6, and the Supplementary figures require enhancement for clarity. The current versions of these plots lack sufficient resolution, making them difficult to read in both print and online formats. Additionally, the font size within these figures should be increased to facilitate better readability.

·         In Figure 5d, the plot title incorrectly mentions 'C57'. This should be corrected to 'WT'. 
